# Integrated Basic Heart and Lung Ultrasound Examination for the Differentiation between Bacterial Pneumonia and Lung Neoplasm in Dogs—A New Diagnostic Algorithm

**DOI:** 10.3390/ani12091154

**Published:** 2022-04-29

**Authors:** Andrzej Łobaczewski, Michał Czopowicz, Agata Moroz, Marcin Mickiewicz, Rafał Sapierzyński, Sylwia Tarka, Tadeusz Frymus, Wojciech Mądry, Michał Buczyński, Olga Szaluś-Jordanow

**Affiliations:** 1Veterinary Clinic Auxilium, Arkadiusz Olkowski, Królewska Str. 64, 05-822 Milanówek, Poland; alobaczewski007@gmail.com; 2Division of Veterinary Epidemiology and Economics, Institute of Veterinary Medicine, Warsaw University of Life Sciences-SGGW, Nowoursynowska Str. 159c, 02-776 Warsaw, Poland; michal_czopowicz@sggw.edu.pl (M.C.); agata_moroz@sggw.edu.pl (A.M.); marcin_mickiewicz@sggw.edu.pl (M.M.); 3Department of Pathology and Veterinary Diagnostic, Institute of Veterinary Medicine, Warsaw University of Life Sciences-SGGW, Nowoursynowska Str. 159, 02-776 Warsaw, Poland; rafal_sapierzynski@sggw.edu.pl; 4Department of Forensic Medicine, Medical University of Warsaw, Oczki 1 Str., 02-007 Warsaw, Poland; sylwia.tarka@wum.edu.pl; 5Department of Small Animal Diseases with Clinic, Institute of Veterinary Medicine, Warsaw University of Life Sciences-SGGW, Nowoursynowska Str. 159c, 02-776 Warsaw, Poland; tadeusz_frymus@sggw.edu.pl; 6Department of Cardiac and General Pediatric Surgery, Medical University of Warsaw, Żwirki i Wigury 63A Street, 02-091 Warsaw, Poland; madwoj1@onet.eu (W.M.); mbuczynski2@wum.edu.pl (M.B.)

**Keywords:** classification and regression trees, dyspnea, pulmonary disease, small animals, thorax

## Abstract

**Simple Summary:**

Dyspnea is a highly alarming sign both for dog owners and veterinarians. Although its recognition is usually easy due to prominent suffering of an animal, finding its cause is challenging because many diseases of the heart, lungs, and airways may manifest themselves this way. Echocardiography and lung ultrasound allow for relatively quick and accurate identification of heart diseases. Dyspneic dogs without a heart and upper airway disease are usually suspected of either bacterial pneumonia or lung neoplasm. Although prognosis in these two conditions is diametrically different, differentiation between them is challenging. Chest radiography is performed in a lateral position, which is barely tolerated by a dyspneic dog, and intensive chest movements often make X-ray scans inconclusive. Computed tomography, although much more accurate, requires general anesthesia, which is difficult and potentially life-threating in a dyspneic dog. Therefore, lung ultrasound, which can be performed quickly in a conscious dog, standing or in sternal position, seems to be the method of choice. We develop and evaluate a diagnostic algorithm based on detection of three well-defined abnormalities in the lung ultrasound. The algorithm allows one to distinguish between bacterial pneumonia and lung neoplasm in a dyspneic dog with high probability of a conclusive result (91%) and high accuracy (>95%).

**Abstract:**

The diagnostics of two of the most prevalent lung diseases in dogs, bacterial pneumonia (BP) and lung neoplasm (LN), are challenging as their clinical signs are identical and may also occur in extrapulmonary diseases. This study aims to identify ultrasonographic criteria and develop a lung ultrasound (LUS)-based diagnostic algorithm which could help distinguish between these two conditions. The study is carried out in 66 dyspneic dogs in which a heart disease was excluded using echocardiography. Based on imaging and laboratory diagnostic tests, as well as follow-up, the dogs are classified into LN (35 dogs) and BP (31 dogs) groups. LUS is performed at admission and the presence of seven lung abnormalities (pleural thickening, B-lines, subpleural consolidations, hepatization with or without aeration, nodule sign and mass classified together as a tumor, and free pleural fluid) and classification and regression trees are used to develop an LUS-based diagnostic algorithm. Distribution of all LUS abnormalities except for aerations differs significantly between groups; however, their individual differentiating potential is rather low. Therefore, we combine them in an algorithm which allows for definitive classification of 60 dogs (91%) (32 with LN and 28 with BP) with correct diagnosis of LN and BP in 31 dogs and 27 dogs, respectively.

## 1. Introduction

The diagnostics of lung diseases are challenging as their main clinical signs, such as coughing or dyspnea, are not specific for any particular disease of the lungs and may also occur in various extrapulmonary diseases such as congestive heart disease and diseases of the upper airways [1]. The introduction of thorax radiography (TXR) to human medicine at the turn of 19th and 20th century was an undeniable breakthrough [2]; however, gradual development of more technically sophisticated methods has revealed its limitations in diagnosing lung diseases [3,4], and prompted development of other imaging modalities. Particularly, lung ultrasound (LUS) examination became a noninvasive and environment-friendly alternative. LUS has high accuracy in the differential diagnostics of pulmonary conditions, both in humans and small animals [5,6]. It has been shown to be more accurate than TXR in diagnosing pneumonia and lung tumors in children and adults [7,8] and plays a significant complementary role to computed tomography (CT), especially being highly sensitive and specific in imaging neoplastic infiltration of the chest wall [9,10,11,12].

According to this knowledge, the question is: why is TXR still widely and easily accepted in pulmonology whereas LUS is not? One possible answer could be that a more accurate diagnostic algorithm for different lung diseases in human medicine and small animal practice is needed. In contrast to abdominal ultrasound (AUS), the presence of air in the respiratory system precludes ultrasonographic visualization of the lungs. However, in pathological processes such as inflammation, edema, neoplasm, embolism, or contusion, the lung tissue is losing air which is replaced by fluid and so-called consolidations become apparent in the ultrasound image. The vast majority of pneumonia lesions in adults have pleural contact [7,13]. Although without the clinical context LUS has limited diagnostic value [14], this limitation applies to all other imaging diagnostic methods [15]. 

LUS is based on the visualization of artifacts such as the pleural line, sliding sign, and A-lines [6,16]. These three artifacts, with a few exceptions, indicate a correct air-filled lung (“dry lung”). The B-lines are the best-known artifacts occurring in the course of lung contusion, pulmonary edema (PE) (cardiogenic and non-cardiogenic), pneumonia, and lung neoplasm [6,17]. These vertical, well-defined lines start from the pleural line and stretch down to the bottom of the screen. In patients with a significant loss of lung aeration, the lines are wider, more numerous, and in a highly advanced stage they merge, forming the picture of so-called “wet lung” [18]. Other artifacts found in pulmonary diseases include various manifestations of lung consolidation like the shred sign, tissue-like sign, and nodule sign. Subpleural consolidations are a diffuse type of lesions, characterized by the presence of numerous, small (2 to 10 mm), solid, focal lesions adjacent to the pulmonary pleura. Shred sign is an area of consolidation which has an irregular border of the lesion. A tissue-like sign is defined as the presence of an area of consolidation of the lung tissue that is adjacent to the pleural line. This ultrasound image is similar in echogenicity and echostructure to the image of the liver. The nodule sign is a round or oval homogeneous, relatively anechoic (or with low echogenicity) area located subpleurally. 

Based on the presence of the aforementioned artifacts, a Bedside Lung Ultrasound in Emergency (BLUE) diagnostic decision tree has been proposed for human patients with severe dyspnea [6,19]. This algorithm links the presence of individual artifacts with the possibility of various diseases. B-lines occur mainly in PE, pneumonia, lung neoplasm, and lung contusion after trauma. In human medicine, subpleural consolidations and irregular pleural line have been shown to occur both in patients with pneumonia and lung neoplasms (70–80% and 45–50%, respectively). Air bronchogram, however, was found in 70% of patients with pneumonia and only 34% of patients with lung neoplasms [20]. Recently, LUS_score_ has been developed as a sum of points describing the occurrence of B-lines for each of eight standardized thoracic locations. It may help to distinguish between lung neoplasms (values < 5) and cardiogenic PE or pneumonia (values ≥ 5) in dogs; however, its accuracy is only fair to moderate [17]. In addition, in human medicine, it is difficult to distinguish between pneumonia and PE based on LUS alone. So far, published data show that using LUS alone leads to the majority of pneumonia cases with B-lines in LUS being misdiagnosed as PE [18]. Therefore, combining LUS results with echocardiography is more useful in this regard [21,22]. LUS with basic cardiac ultrasound is widely used in human medicine under the name of thoracic ultrasound (TUS) [23,24]. Furthermore, in veterinary medicine, a basic cardiac examination is a very useful tool [25]. TUS in patients with severe heart disease and multiple B-lines strongly indicates cardiogenic PE whereas an unchanged heart in basic echocardiography rules it out. On the other hand, the ultrasonographic view of conditions causing dyspnea which are unrelated to heart disease such as pneumonia and lung neoplasm appears to be similar [26]. As systematic investigations of the discriminatory potential of LUS in these two diseases are lacking, our study aims to identify ultrasonographic criteria and develop a diagnostic algorithm which could help distinguish between pneumonia and lung neoplasm in dyspneic canine patients.

## 2. Materials and Methods

### 2.1. Animals

The study was carried out between 2016 and 2021 and all adult dogs (at least 12-month-old) admitted in a round-the-clock veterinary clinic for dyspnea were enrolled. For each dog, the following demographic and clinical data were recorded: breed, gender, age, body weight, history of the disease, and medications used during the last month. Standard clinical examination was followed by an ultrasound of the heart and lungs. On this basis, an initial diagnosis was made and relevant therapy was started.

According to the Polish legal regulations (The Act of the Polish Parliament of 15 January 2015 on the Protection of Animals Used for Scientific or Educational Purposes, Journal of Laws 2015, item 266), the consent of the ethics committee was not required as the study only involved standard procedures essential for appropriate treating of patients and postmortem tissue use. All owners granted informed consent for the inclusion of their animals in the study. 

### 2.2. Heart and Lung Ultrasound Examination

Heart and lung ultrasound examinations were performed using three ultrasound devices: GE Healthcare Logiq F6 (Waukesha, WI, USA) with a 10-6 MHz microconvex transducer and 13-6 MHz linear transducer; Mindray M7 with a 4-2s MHz phased array transducer, 12-4s MHz phased array transducer, and L14-6ns MHz linear transducer (Shenzhen 518057, China); Mindray M9 (Shenzhen 518057, China) with a 5-1s MHz phased array, 10-4s MHz phased array, and L14-6ns MHz linear transducer. Microconvex or phased array transducers were used for basic heart examination and microconvex or linear transducers were used for lung examination [17]. Ultrasonographic examinations were performed by two board-certified specialists in TUS (OSJ and AŁ), each with over 10 years of experience in this field. 

Dogs were examined in standing or sternal positions. The hair was only parted without clipping and an appropriate amount of alcohol and coupling gel was used. TUS examination was performed according to the previously reported guidelines [6,17,27].

### 2.3. Inclusion and Exclusion Criteria

Dogs were included in the study if the following 3 criteria were simultaneously satisfied:Presence of dyspnea, defined as respiratory rate > 30 bpm and subjective exercise intolerance reported by the owner and confirmed by the veterinary staff during the consultation.Dimensions and function of the heart and cardiac chambers within reference ranges [28]. No excess fluid in the pericardial sac and no active inflammatory or neoplastic changes in the heart.Presence of at least one of the following abnormalities found in LUS:
Irregular and thickened pleural line, henceforth referred to as pleural thickening (Figure 1); B-lines (Figure 2); Subpleural consolidations (Figure 3 and Figure 4); Translobar consolidation producing tissue-like sign, henceforth referred to as hepatization (Figure 5, Figure 6 and Figure 7); Pulmonary nodules (≤3 cm in diameter), visualized as an anechoic or hypoechogenic area adjacent to the pleural line of round or oval shape, often with visible B-lines extending from its lower edge (Figure 8), and pulmonary masses (>3 cm in diameter), usually appearing as hypoechogenic masses with a clear-cut deep margin (Figure 9). Pulmonary nodules and masses were merged into one category, henceforth referred to as tumor;Aerations visualized as air entrapment in the abnormally modified lung tissue which usually accompanies consolidation or hepatization (Figure 7);Free pleural fluid (Figure 10).

The presence of normal, unchanged pleural line and A-lines with a lung sliding was considered as normal, aerated lung, and all these patients were excluded from the further study as suspected of having dyspnea originating from the upper airways or chronic bronchitis. 

Patients suspected of pneumonia based on LUS were put on a 2-week broad-spectrum antibiotic treatment (ceftiofur, Excenel Fluid, Zoetis, Poland and marbofloxacin, Marbocyl, Vetoquinol Biowet, Poland) and a follow-up LUS was performed 2 weeks after the initiation of treatment. If normal aerated lungs were visualized and the dog’s clinical condition reverted to normal, the treatment was discontinued. Otherwise, the treatment was maintained until the next follow-up LUS, 1 week later, revealed normal aerated lung. The definitive diagnosis of bacterial pneumonia (BP) was established based on positive response to the treatment, analogically to a large-scale prospective study on PE and pneumonia carried out recently in human medicine [20]. 

Patients suspected of lung neoplasia underwent chest radiography (27 dogs) or chest CT-scan (5 dogs) together with TUS. To confirm the neoplastic disease, the patients underwent ultrasound-guided transthoracic fine-needle aspiration lung biopsy or thoracocentesis and free pleural fluid aspiration, and the material was referred to the pathology laboratory for further cytological examination.

### 2.4. Final Classification of Patients

A dog was allocated in the lung neoplasm (LN) group if: (i) thoracic radiography or CT reviewed by a board-certified radiologist was indicative of LN or (ii) microscopic examination of the free thoracic fluid or the abnormal lung performed by a board-certified veterinary pathologist revealed neoplastic cells. 

LN was classified as: (i) primary (Figure 11), if the histopathological examination identified the tumor as primary LN and no tumor was detected in any other organ in the clinical examination and AUS, (ii) likely metastatic (Figure 12), if the malignant tumor was found in any other organ, or (iii) of unknown origin, if microscopic verification of the lung lesions was lacking and no neoplastic lesions were found in any other organs in clinical examination or in the following auxiliary tests: complete blood count and biochemistry blood panel, urine check-up, AUS, and TXR.

A dog was allocated in the bacterial pneumonia (BP) group if: (i) dyspnea resolved after a broad-spectrum antibacterial treatment and the follow-up LUS after 2–3 weeks revealed resolution of LUS abnormalities.

Dogs were excluded if: (i) the clinical history indicated lung contusion, hemorrhage, or (ii) dyspnea was identified to originate from the upper airways obstruction; or (iii) clinically significant heart disease and PE or heart neoplasm was diagnosed in TUS.

### 2.5. Statistical Analysis

Numerical variables were expressed as the median, interquartile range (IQR) and range, and compared between groups using the Mann–Whitney U test. Categorical variables were presented as counts and proportions, and compared between groups using the maximum likelihood G-test or Fisher exact test if the count of any cell in the contingency table was <5. The 95% confidence intervals (CI 95%) for proportions were calculated using the Wilson score method [29]. All tests were 2-tailed. A significance level (α) was set at 0.05. Statistical analysis was performed in TIBCO Statistica 13.3 (TIBCO Software Inc., Palo Alto, CA, USA)

To select LUS abnormalities which could help distinguish between LN and BP, their link to the diagnosis was evaluated at two levels. First, the proportion of animals with a given LUS abnormality was compared between dogs with LN and BP using the maximum likelihood G-test or Fisher exact test as described above. Secondly, the number of quadrants (from 0 to 8) in which a given LUS abnormality had been detected was compared between dogs with LN and BP using the Mann–Whitney U test. If the latter was significant at α = 0.05, the ROC analysis was performed and the area under ROC curve (AUROC) was used to classify discriminatory potential of the number of quadrants affected as high (AUROC > 90%), moderate (>80% to 90%), fair (>70% to 80%), or poor (≤70%) [30,31]. The optimal cut-off value (i.e., the number of quadrants affected above which LN or BP was more likely) was identified by the maximization of the Youden’s index (J), given by the formula:(1)J=Se−(1−Sp)
where Se was a diagnostic sensitivity and Sp was a diagnostic specificity. If J index differed between the categorization of dogs as LUS abnormality present vs. LUS abnormality absent and categorization according to the number of quadrants affected by LUS abnormality, the categorization associated with the higher J index was chosen.

Subsequent analysis was carried out using the Classification and Regression Trees (CART), a non-parametric supervised learning technique, built according to the trimming algorithm in the Data Mining module of TIBCO Statistica 13.3 (TIBCO Software Inc., Palo Alto, CA, USA). The classification was done by maximizing the information gain (IG), which showed the difference in impurity between the parent (decision) node (Gini impurity of the parent node [I_Gini_(D)]) and child nodes (weighted sum of Gini impurity of child nodes [I_Gini_(D_k_)]) and was given by the equation: (2)$$\mathrm{IG}\left(D_{p},f\right)=I_{\mathrm{Gini}}\left(D_{p}\right)-\sum_{1}^{2}\frac{n_{k}I_{\mathrm{Gini}}\left(D_{k}\right)}{n}=1-\left(p_{\mathrm{LN}\left(p\right)}^{2}+p_{\mathrm{BP}\left(p\right)}^{2}\right)-\sum_{1}^{2}\frac{\left(1-\left(p_{\mathrm{LN}\left(k\right)}^{2}+p_{BP\left(k\right)}^{2}\right)\right)n_{k}}{n}$$
where f was a feature to perform the split (i.e., a particular LUS abnormality), D_p_ was a data set in the parent node, D_k_ with k = {L,R} was a data set in the left (L) or right (R) child node, n was the number of observations in the parent node, n_k_ with k = {L,R} was the number of observations in the left (L) or right (R) child node, and p_LN(k)_ and p_BP(k)_ with k = {L,R} were the proportion of observations from class LN or BP, respectively, in the left (L) and right (R) child node. I_Gini_(D) was the impurity function (function describing distribution of the same elements to 2 classes) based on the Gini index (GI) and given by the formula:(3)$$I_{\mathrm{Gini}}\left(D\right)=1-\mathrm{GI}=\sum_{i=1}^{2}p_{i}\left(1-p_{i}\right)=1-\sum_{i=1}^{2}p_{i}^{2}=1-\left(p_{\mathrm{LN}}^{2}+p_{\mathrm{BP}}^{2}\right)$$
where p_i_ was a fraction of data of class i with i = {LN,BP} in a subset D. I_Gini_(D) could take values from 0 (the best purity—all animals in a leaf node belong to a single class) to 0.50 (the worst purity—animals in a leaf node are split half and half).

The pre-test (prior) probability of LN (pre-P_LN_) and BP (pre-P_BP_) was set as equal (i.e., 50%). The CART was built assuming an equal cost of misclassification and splitting was stopped when the count at decision (parent, internal) node was ≤10. Positive (LR+) and negative (LR-) likelihood ratios, with CI 95% calculated using the log method [29], were used for evaluation of the magnitude of diagnostic potential of a given LUS abnormality. By default, a diagnosis of LN was denoted as a positive result (i.e., indicated by LR+) and a diagnosis of BP was denoted as a negative result (i.e., indicated by LR-). Post-test probability of LN (post-P_LN_) and BP (post-P_BP_) were calculated by converting probabilities into odds and multiplying by LR+ in terms of LN or LR- in term of BP, analogically to calculating positive and negative predictive value using the Bayesian approach [17,32,33].
(4)$${\mathrm{postP}}_{\mathrm{LN}}=\frac{\frac{{\mathrm{preP}}_{\mathrm{LN}}}{1-{\mathrm{preP}}_{\mathrm{LN}}}\times {\mathrm{LR}}_{+}}{1+\frac{{\mathrm{preP}}_{\mathrm{LN}}}{1-{\mathrm{preP}}_{\mathrm{LN}}}\times {\mathrm{LR}}_{+}}$$
(5)$${\mathrm{postP}}_{\mathrm{BP}}=\frac{\frac{{\mathrm{preP}}_{\mathrm{BP}}}{1-{\mathrm{preP}}_{\mathrm{BP}}}\times {\mathrm{LR}}_{-}}{1+\frac{{\mathrm{preP}}_{\mathrm{BP}}}{1-{\mathrm{preP}}_{\mathrm{BP}}}\times {\mathrm{LR}}_{-}}$$

## 3. Results

The study population comprised 66 adult dogs (35 males and 31 females), aged from 1 to 17 years with a median (IQR) of 11 (9–13) years. Thirty-five dogs (53.0%) had lung neoplasm (LN group) and 31 (47.0%) had bacterial pneumonia (BP group). Neither age nor sex distribution differed significantly between groups (Table 1). Most dogs from the BP group were pedigree compared to only half in the LN group (*p* = 0.021). 

Of the 35 dogs from the LN group, 20 (57.1%) had metastatic LN from testis, liver, spleen (three dogs each), mammary gland (two dogs), thyroid gland, prostate, stomach, unknown glandular organ, skin, kidney, mediastinum, and bone—scapula (one dog each); one dog had pulmonary leukemic infiltrates. Nine dogs (25.7%) had primary LN—bronchoalveolar carcinoma (five dogs), adenocarcinoma (three dogs), and histiocytosis (one dog). In six dogs (17.2%), the origin of LN was unknown. 

B-lines, subpleural consolidations, and pleural thickening were the most prevalent LUS abnormalities found in 78.8% (CI 95: 67.5–86.9%; n = 52), 72.7% (CI 95: 61.0–82.0%; n = 48), and 65.2% (CI 95: 53.1–75.5%; n = 43) of dogs, respectively. Tumor and hepatization were observed in roughly one third of dogs—37.9% (CI 95: 27.1–49.9%; n = 25) and 31.8% (CI 95: 21.8–43.8%; n = 21), respectively, whereas free pleural fluid and aerations in roughly one fourth of dogs—25.8% (CI 95: 16.7–37.4%; n = 17) and 22.7% (CI 95: 14.3–34.2%; n = 15), respectively. Distribution of all abnormalities except for aerations differed significantly between groups (Figure 13, Table 2). B-lines (*p* = 0.027), subpleural consolidations (*p* = 0.012), and pleural thickening (*p* = 0.012) were significantly more common in the BP group, whereas tumor (*p* < 0.001), hepatization (*p* = 0.001), and free pleural fluid (*p* = 0.004) were more common in the LN group.

A significantly higher number of quadrants affected by a LUS abnormality was observed in dogs in the BP group with respect to B-lines (AUROC = 66.3%; CI 95%: 51.8–78.8%; *p* = 0.026), subpleural consolidations (AUROC = 73.7%; CI 95%: 61.0–86.5%; *p* < 0.001), and pleural thickening (AUROC = 69.7%; CI 95%: 56.5–82.8%; *p* = 0.003) (Figure 14). The optimal cut-off values were set at ≥3 quadrants affected, indicating BP for all three abnormalities and classification based on these cut-off values was better according to the J index in terms of all three LUS abnormalities (Table 2). Therefore, this form of categorization of dogs according to the aforementioned LUS abnormalities was included in the CART model. Tumor (AUROC = 85.7%; CI 95%: 76.1–95.3%; *p* < 0.001), hepatization (AUROC = 67.1%; CI 95%: 53.8–80.3%; *p* = 0.011), and free pleural fluid (AUROC = 65.4%; CI 95%: 52.2–78.6%; *p* = 0.022) were observed in the significantly higher number of quadrants of dogs in the LN group (Figure 14). The optimal cut-off values were set on ≥1 quadrants (i.e., any quadrant) affected indicating LN for these three abnormalities (Table 2), and they were included in this form in the CART model. The presence of a tumor in LUS was the only abnormality whose sole AUROC had at least fair diagnostic accuracy (the entire CI 95% > 70%). For the rest of the LUS abnormalities which significantly differed between the two groups of dogs, diagnostic accuracy was poor (the lower bound of CI 95% < 70%).

The distribution of LN and BP in the study population was not significantly different from 1:1 (*p* = 0.491), so the pre-test probability of LN (pre-P_LN_) was assumed to be 50%. The CART flowchart is shown in Figure 15.

A tumor was the first decision (internal) node in CART (Figure 15—I). Detection of a tumor in LUS was associated with LR+ > 99, so the estimated post-test probability of LN (post-P_LN_) was >99%. The observed post-P_LN_ was 100% (CI 95%: 86.7–100%; 25/25 dogs). Therefore, detection of a tumor was considered as a leaf node (i.e., definitive decision) of CART corresponding to a positive result of LUS for LN. A LUS negative for a tumor was associated with LR- = 0.29, so the estimated post-P_LN_ was 22.5%. The observed post-P_LN_ was 24.4% (CI 95%: 13.8–39.3%; 10/41 dogs). Therefore, no tumor in the LUS was considered as an inconclusive result and the diagnostic procedure carried on.

The second decision (internal) node in CART was hepatization (Figure 15—II). LUS being negative for hepatization was associated with LR- = 0.12, so the estimated post-P_LN_ was only 3.4% and the observed post-P_LN_ was 3.6% (CI 95%: 0.6–17.7%; 1/28 dogs). As post-P_BP_ was 96.4% (CI 95%: 82.3–99.4%), the parallel combination of negative results for tumor and hepatization was considered as the second leaf node of CART corresponding to a positive result of LUS for BP. A LUS positive result for hepatization in a dog negative for a tumor was associated with LR+ = 7.0, so the estimated post-P_LN_ was 67.0% and the observed post-P_LN_ was 69.2% (CI 95%: 42.4–87.3%; 9/13 dogs). The result was considered inconclusive, and the diagnostic procedure carried on.

The third decision (internal) node in CART was subpleural consolidation (Figure 15—III). Detection of subpleural consolidations in <3 quadrants was associated with LR+ = 2.7, so the estimated post-P_LN_ was 84.6% and the observed post-P_LN_ was 85.7% (CI 95%: 48.7%–97.4%; 6/7 dogs). Therefore, the detection of subpleural consolidations in <3 quadrants in a dog negative for tumor but positive for hepatization in LUS was considered as a leaf node of CART, corresponding to a positive result of LUS of LN.

Detection of subpleural consolidations in ≥3 quadrants was associated with LR- = 0.44, so the estimated post-P_LN_ was 47.2% (post-P_BP_ = 52.8%) and the observed post-P_LN_ was 50.0% (CI 95%: 18.8–81.2%; 3/6 dogs). Therefore, the result was considered inconclusive. The diagnostic procedure was discontinued at this step as no further refinement of the classification could be obtained using LUS abnormalities.

The procedure led to a conclusive result in 60 of 66 dogs (90.9%; CI 95%: 81.6–91.8%)—32 positive for LN and 28 positive for BP. Six dogs (9.1%—three with LN and three with BP) were classified as inconclusive. LN diagnosis was correct in 31 of 32 LN dogs with a conclusive result (96.9%; CI 95%: 84.3–99.4%) and BP diagnosis was correct in 27 of 28 dogs with a conclusive result (96.4%; CI 95%: 82.3–99.4%). 

## 4. Discussion

Our study evaluated the capability of a set of LUS abnormalities in distinguishing between two very common causes of lung-associated dyspnea—(BP) and (LN). Even though the prevalence of six of seven LUS abnormalities differed significantly between the two groups of dyspneic dogs, the magnitude of individual differences was small (i.e., AUROC was low for most of them). Only the presence of a tumor in LUS was associated with individual diagnostic accuracy that was sufficient from the clinical standpoint (i.e., AUROC > 70%) [30,31]. Therefore, we decided to combine selected LUS abnormalities into a single diagnostic algorithm. We chose a model based on the Classification and Regression Trees (CART) as this method allows for gradual incorporation of subsequent elements and can be stopped at various stages if the amount of evidence in favor of a particular disease is sufficient. Thanks to this, we could develop a practically useful and convenient diagnostic algorithm.

Dyspnea is a life-threatening clinical sign associated with high mortality unless properly treated. An appropriate and timely treatment can only be applied if an accurate diagnosis has been made as soon as possible. Many scientific publications have shown that LUS is now an important and accurate diagnostic test for respiratory diseases [6,17,34]. TUS significantly extends the usefulness of LUS for a comprehensive assessment of the circulatory and respiratory status [18,23]. Based on echocardiography and the presence of B-lines in LUS, the most common cardiogenic cause of dyspnea, PE, can be accurately confirmed, which immediately leads to a definitive diagnosis in a great proportion of dyspneic dogs [17,34,35,36]. If TUS is negative for cardiogenic PE, the clinician needs to distinguish between a disease of the lungs and of the upper airways. LUS is also helpful at this stage as the lack of any abnormalities indicates that dyspnea originates from the upper airways or bronchitis [6]. The most problematic is a dyspneic dog with normal heart and abnormalities in LUS, indicative of the pulmonary origin of dyspnea. This is the kind of patient we focused on in this study. The two conditions usually suspected in these patients is LN and BP. Both are life-threatening; however, they differ diametrically with respect to the prognosis, which is at least fair in BP and at most poor in LN [37].

In this study, we focused on pneumonia which can be classified as of bacterial etiology, because definitive diagnosis was based on clinical criteria, mainly recovery on a broad-spectrum antibiotic treatment. There is, however, one important pitfall of such an approach. Pneumonia does not have to be of bacterial etiology. Although it appears to be most prevalent in the canine population [38], viral, fungal, and parasitic pneumonias are also possible and there is no basis to claim that our diagnostic algorithm will work equally well in such cases. Nevertheless, we think that our study still provides very important and useful results, because the main question asked in a dyspneic dog with normal heart and abnormalities in LUS indicative of the pulmonary origin of dyspnea is whether it has lung neoplasia or not. If the answer is positive, humane euthanasia should at least be carefully considered due to the dog’s suffering and poor prognosis. Otherwise, aggressive treatment is indicated as the dog’s life may likely be saved. Our diagnostic algorithm may serve as a useful auxiliary tool in making this difficult decision.

We decided to express LUS abnormalities as dichotomous (binary) variables for two reasons. First, to make the classification of patients with respect to these abnormalities as intuitive and simple as possible. In clinical conditions, including the exact number of lung quadrants affected by a given abnormality in the model would consume time, bring about mistakes, and eventually discourage clinicians from using the diagnostic algorithm. Secondly, dichotomous categorization of a numerical variable made the algorithm more robust because a small change in the number of quadrants affected by the abnormality, which is likely to result from imperfect repeatability and reproducibility of LUS, is unlikely to influence the final patient classification.

However, the number of quadrants in which LUS abnormalities could be detected may be considered as a proxy of severity and dissemination of a given pathological process in the lungs. As binary categorization of a numerical variable leads to loss of some amount of information, the method that retains maximum information should be chosen. We used methods commonly used in epidemiology (ROC analysis and maximization of Youden’s index) to search for a better classification than the method based on simple “present vs. absent” categorization. They allowed for the dichotomization of three of the LUS abnormalities (B-lines, subpleural consolidations, and pleural thickness) in a manner that separated dogs with LN from dogs with BP more accurately than when a “present vs. absent” method was used. Therefore, we decided to enter them in the CART model in this form.

The CART model was finally based on three of six initially included LUS abnormalities—the presence of tumor, hepatization, and less than three vs. at least three quadrants with subpleural consolidations. The first decision node, i.e., the moment for classification of patients in our CART model, was the detection of at least one tumor in LUS. According to our results, such a patient was very likely to have LN as all patients with a tumor in LUS had LN. However, given the limited number of dogs enrolled in our study, it is much safer to claim that the probability of LN in such a patient is at least 85% (lower confidence limit of the observed post-test probability). Anyway, detection of a tumor in LUS is the first conclusive result of our diagnostic algorithm. Subpleurally localized tumors appear as hypoechoic masses, usually with a more clearly marked border from the normal lung, compared to inflammatory consolidations. Most often the masses are round, polypoid, or triangular and less often they irregularly penetrate the normal lung tissue. Contrary to inflammatory condensations, the shape and size of neoplastic lesions do not change over a short period of time. The echogenicity of lung tumors is variable, depending on the type of tumor and its size. It is not possible to diagnose the type of tumor on the basis of the LUS image. Aeration may be also present in the tumor tissue, resulting from air entrapment in the alveoli or bronchi [39]. 

However, the lack of a tumor in LUS does not exclude LN—roughly one fourth (10/41) of dogs without the tumor in LUS in fact had LN. Therefore, the next decision node was the detection of hepatization, i.e., altered lung tissue that resembled liver tissue on the image. In this situation, the conclusive result was the lack of hepatization in any of eight quadrants of the lungs. If a dyspneic dog did not have either a tumor (as stated in the previous paragraph) or hepatization, it was very likely (96%) to have BP. Again, having taken into account a relatively small number of dogs in our study, it is safe to claim that the probability of BP in such a patient is at least 82% (lower confidence limit of the observed post-test probability). 

A dyspneic dog without a tumor but with hepatization in LUS still remained a puzzle. In our study, such a patient was more likely to have LN, but the probability was not significantly different from the probability of BP (ratio of roughly 2:1). Therefore, we searched for the next decision node. It turned out to be the presence of subpleural consolidations in less than three quadrants. Such a result in a dog negative for tumor but positive for hepatization was conclusive and indicated LN. It must be stressed that at this level of the analysis the number of dogs was already very low—only seven—out of which one dog had BP and six dogs had LN. Although the expected and observed post-test probabilities of LN were roughly 85%, the precision of this estimation was very low (CI 95% was very wide), so this result should be treated with very high caution. Especially when the diagnosis of LN is going to constitute a basis for euthanasia, another diagnostic method should be used to attain higher confidence of the diagnosis. The weakness of this last decision node is emphasized by the CI 95% for LR+ covering 1, which in fact indicates that such a result may also be obtained in a dog with BP. Regardless, given the fact that the diagnostics of intrathoracic pathological processes are very challenging in veterinary medicine, we think that even such a weak conclusion deserves some attention, at least as a starting point for further studies. 

A dyspneic dog without a tumor in LUS, with hepatization in LUS, and with subpleural consolidations in ≥3 quadrants was equally likely to have LN and BP, so it had to be considered as an inconclusive result of the CART model. The overall discriminative potential of the diagnostic algorithm was high. Based on the lower limit of the CI 95%, it was very likely to yield conclusive results in over 80% of dyspneic dogs and the classification made with its use was more than 80% likely to be true. Given that an alternative in a dyspneic patient is TXR or CT with their multiple limitations, these figures appear to be sufficient to recommend the use of the diagnostic algorithm based on the LUS-CART model in daily veterinary practice. Obviously, it must be stressed once more that this diagnostic algorithm applies to the patient whose dyspnea does not result from a disease of the heart or upper airways or from lung contusion. Due to the relatively small study population, we could estimate diagnostic accuracy measures of our algorithm with only moderate or sometimes even low precision (wide CI 95%). Therefore, our results warrant verification on a larger study population.

## 5. Conclusions

In summary, a diagnostic algorithm based on LUS can serve as an accurate and easily available method of differentiation between bacterial pneumonia and lung neoplasia in a dyspneic dog without pulmonary edema (based on TUS examination), upper airway disease, and lung contusion. 

## Figures and Tables

**Figure 1 animals-12-01154-f001:**
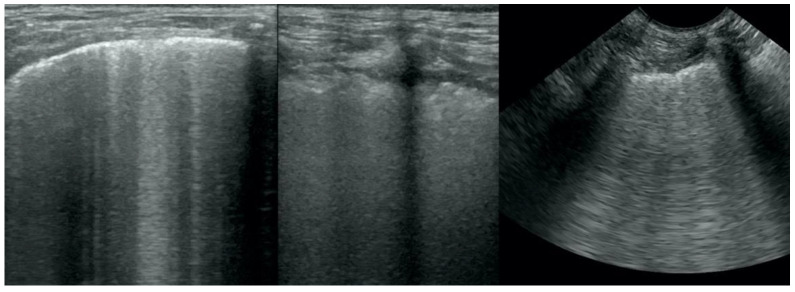
Irregular and thickened pleural line.

**Figure 2 animals-12-01154-f002:**
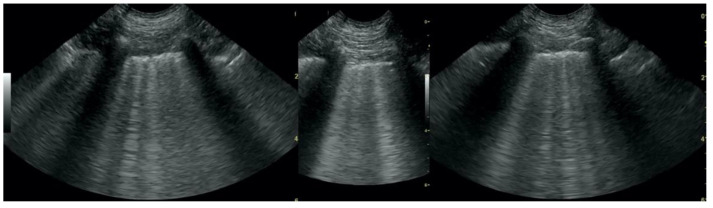
Multiple B-lines.

**Figure 3 animals-12-01154-f003:**
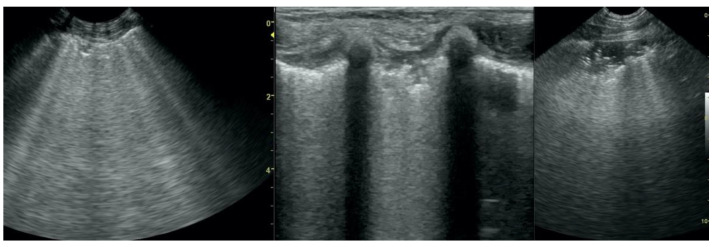
Subpleural consolidations in the course of pneumonia.

**Figure 4 animals-12-01154-f004:**
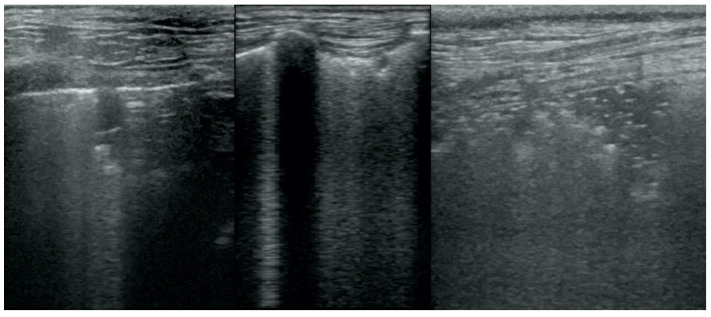
Subpleural consolidations in the course of lung cancer.

**Figure 5 animals-12-01154-f005:**
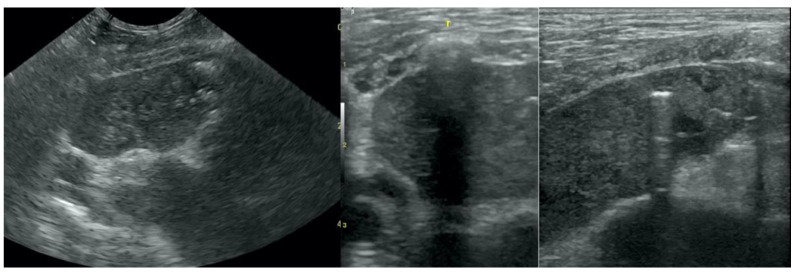
Hepatization without aeration in the course of pneumonia.

**Figure 6 animals-12-01154-f006:**
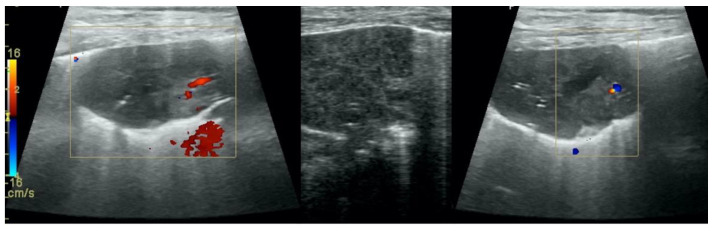
Hepatization without aeration in the course of lung cancer.

**Figure 7 animals-12-01154-f007:**
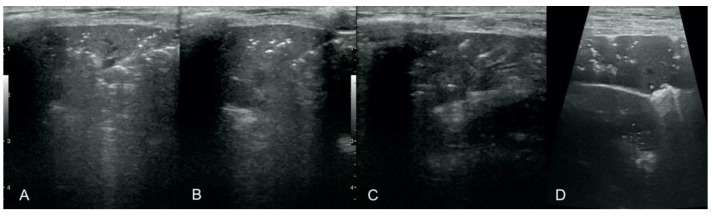
Hepatization with aeration in the course of pneumonia (**A**–**C**) and lung cancer (**D**).

**Figure 8 animals-12-01154-f008:**
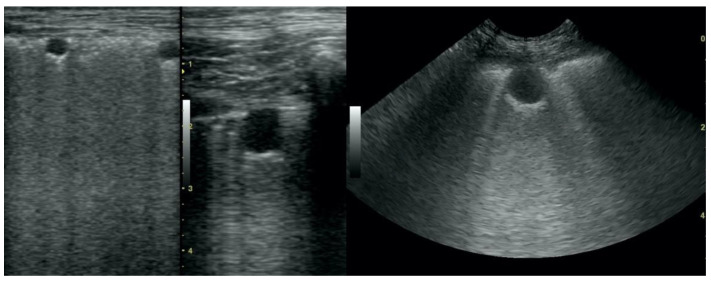
Nodule sign in the course of lung cancer.

**Figure 9 animals-12-01154-f009:**
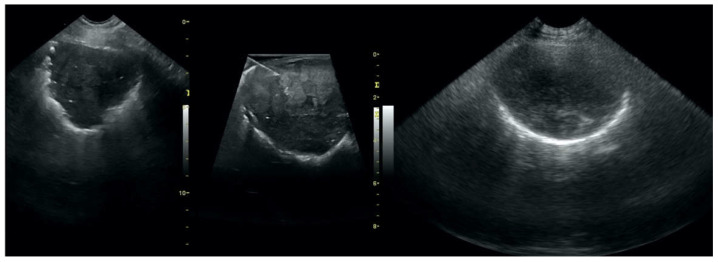
Subpleural mass in the course of lung cancer. At the middle scan, biopsy needle is visible.

**Figure 10 animals-12-01154-f010:**
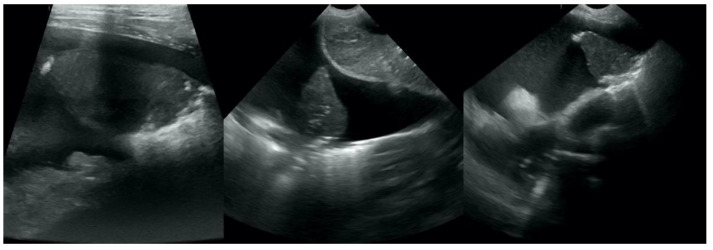
Hepatization in the course of lung cancer and free fluid in thorax cavity.

**Figure 11 animals-12-01154-f011:**
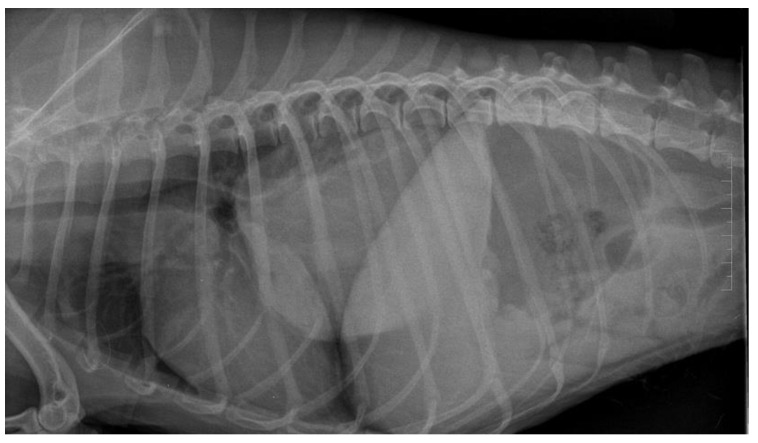
Lateral thoracic radiograph of a dog showing a large round shadow, occupying most of the caudal pulmonary field. The shadow was diagnosed as a primary solitary lung neoplasm in autopsy and was identified as adenocarcinoma in the histopathological examination.

**Figure 12 animals-12-01154-f012:**
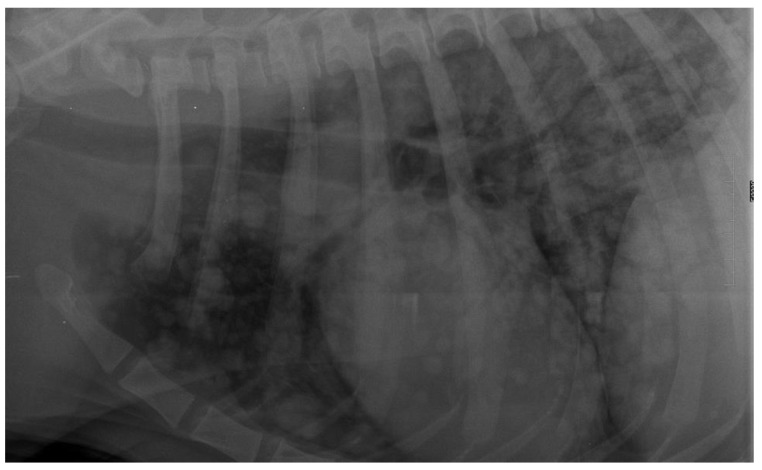
Lateral thoracic radiograph of a dog showing multiple small round shadows scattered in the entire cranial and caudal pulmonary fields. The shadows were diagnosed as a pulmonary metastasis of mammary gland neoplasm in the autopsy and were identified as carcinoma in the histopathological examination.

**Figure 13 animals-12-01154-f013:**
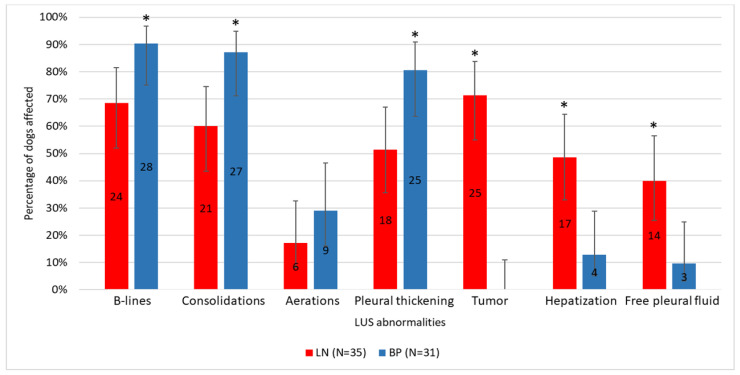
Distribution of lung ultrasound (LUS) abnormalities in dogs with lung neoplasm (LN) and bacterial pneumonia (BP). The number of dogs with a given LUS abnormality in a group is presented in the center of each bar. Whiskers correspond to the 95% confidence interval (CI 95%). Asterisks (*) indicate a group with significantly higher prevalence of a given LUS abnormality according to the maximum likelihood G-test.

**Figure 14 animals-12-01154-f014:**
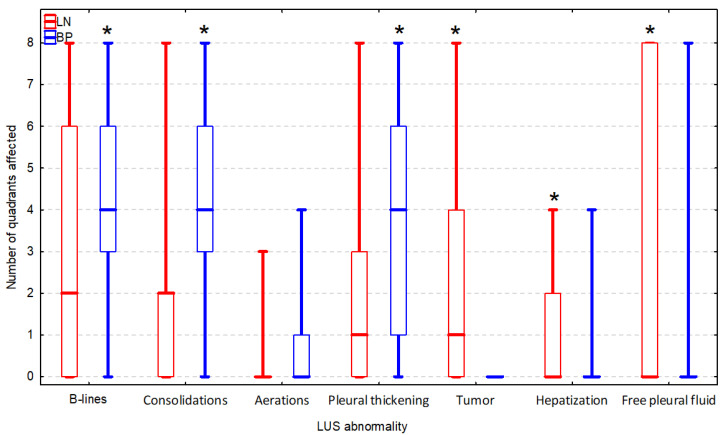
The number of quadrants in which a given lung ultrasound (LUS) abnormality was detected in dogs with lung neoplasm (LN, n = 35) and bacterial pneumonia (BP, n = 31). Bars, boxes, and whiskers correspond to the median, interquartile range (IQR), and range, respectively. Asterisks (*) indicate a group with significantly higher number of quadrants affected by a given LUS abnormality according to the Mann–Whitney U test.

**Figure 15 animals-12-01154-f015:**
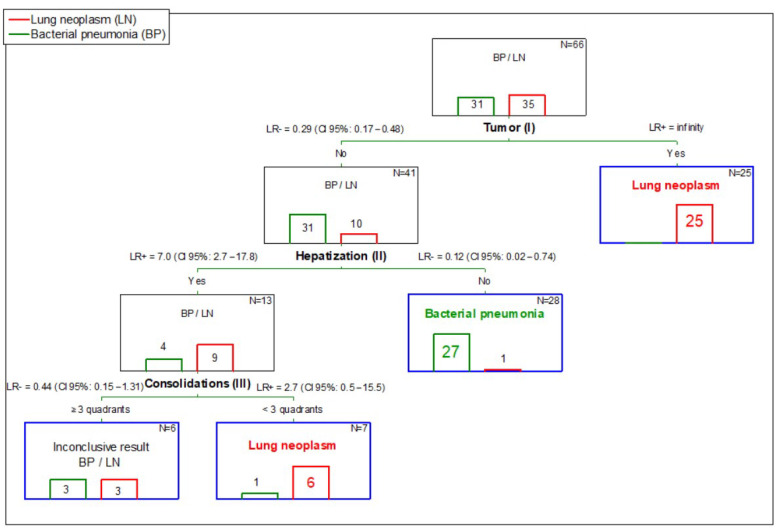
A Classification and Regression Tree (CART) for differentiation between lung neoplasm (LN) and bacterial pneumonia (BP) in dyspneic dogs without heart disease based on three lung ultrasound (LUS) abnormalities—tumor, hepatization, and subpleural consolidations. LR+ and LR- stand for positive and negative likelihood ratio, respectively. CI 95% is a 95% confidence interval.

**Table 1 animals-12-01154-t001:** Demographic characteristic of the study population.

Demographic Characteristics	Lung Neoplasm [LN] (n = 35)	Bacterial Pneumonia [BP] (n = 31)	*p*-Value
Age ^a^	11, 10–13 (6–17)	11, 8–14 (1–17)	0.737
Sex—males ^b^	16 (45.7)	19 (61.3)	0.205
Pedigree ^b^	19 (54.3)	25 (80.7)	0.021
Breed	German shepherd (3), Pointer (3), Poodle (2), dachshund, Boxer, cocker spaniel, Yorkshire terrier, rottweiler, miniature schnauzer, giant schnauzer, Doberman, Bernese mountain dog, whippet, golden retriever (1 each)	Yorkshire terrier (5), French bulldog (3), golden retriever (2), miniature pinscher (2), beagle, German shepherd, whippet, Pekinese, collie, WHWT, cocker spaniel, bloodhound, Staffordshire terrier, Labrador retriever, pug, Shih Tzu (1 each)	

^a^ presented as median, IQR, and range. ^b^ presented as count and percentage.

**Table 2 animals-12-01154-t002:** Comparison between the definitive diagnosis and lung ultrasound (LUS) abnormalities.

LUS Abnormality	The Number (%) of Dogs with		G-Test *p*-Value	Youden’s Index (CI 95%)
	Lung Neoplasm [LN] (n = 35)	Bacterial Pneumonia [BP] (n = 31)		
B-lines				
Any quadrant affected indicates BP	24 (68.6)	28 (90.3) *	0.027	21.8 (12.3–31.2)
≥3 quadrants affected indicate BP	14 (40.0)	24 (77.4) *	0.002	37.4 (26.2–48.6)
Subpleural consolidations				
Any quadrant affected indicates BP	21 (60.0)	27 (87.1) *	0.012	27.1 (16.9–37.3)
≥3 quadrants affected indicate BP	8 (22.9)	24 (77.4) *	<0.001	54.6 (44.2–64.9)
Aerations				
Any quadrant affected indicates BP	6 (17.1)	9 (29.0)	0.250	-
Pleural thickening				
Any quadrant affected indicates BP	18 (51.4)	25 (80.7) *	0.012	29.2 (18.2–40.2)
≥3 quadrants affected indicate BP	9 (25.7)	22 (71.0) *	<0.001	45.3 (34.3–56.3)
Tumor				
Any quadrant affected indicates LN	25 (71.4) *	0	<0.001	71.4 (63.8–79.1)
Hepatization				
Any quadrant affected indicates LN	17 (48.6) *	4 (12.9)	0.001	35.7 (25.3–46.0)
Free pleural fluid				
Any quadrant affected indicates LN	14 (40.0) *	3 (9.7)	0.004	30.3 (20.5–40.2)

* significantly higher at α = 0.05.

## Data Availability

The data sets used and/or analyzed are available from the corresponding author on reasonable request.

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
