# Peer review of "Integrated Basic Heart and Lung Ultrasound Examination for the Differentiation between Bacterial Pneumonia and Lung Neoplasm in Dogs—A New Diagnostic Algorithm"

_animals, 2022, doi:10.3390/ani12091154_

Round 1
Reviewer 1 Report
The manuscript "Integrated basic heart and lung ultrasound examination for the differentiation between bacterial pneumonia and lung neoplasm in dogs – a new diagnostic algorithm" (Manuscript ID: animals-1676040) faces with a methodological approach two important pathologies of dogs, lung neoplasm and bacterial pneumonia.
The manuscript is well structured and clear in all its parts. The only advice I would like to give to the authors is to dedicate a part of the Introduction to the description of the pathologies taken into consideration. In particular, please specify the etiological agents for bacterial pneumonia.
Author Response
The manuscript "Integrated basic heart and lung ultrasound examination for the differentiation between bacterial pneumonia and lung neoplasm in dogs – a new diagnostic algorithm" (Manuscript ID: animals-1676040) faces with a methodological approach two important pathologies of dogs, lung neoplasm and bacterial pneumonia.
Reviewer: The manuscript is well structured and clear in all its parts. The only advice I would like to give to the authors is to dedicate a part of the Introduction to the description of the pathologies taken into consideration.
Our answer: done
Reviewer: In particular, please specify the etiological agents for bacterial pneumonia.
Our answer: In our research, the initial diagnosis of pneumonia was based on a clinical examination, clinical interview combined with an ultrasound image, and the confirmation of the diagnosis was the disappearance of lesions after the applied broad-spectrum antibiotic therapy and the animal's recovery to full health. Unfortunately, we have no confirmed etiological factors in our patients.
Reviewer 2 Report
This manuscript proposes a diagram to differentiate lung neoplasia to pneumonia. However, authors mixed primary and metastatic diseases. Therefore, could be interesting to shown some x-ray images of the disease diseases, to increase reader understanding.
Authors could provide more information in the manuscript to direct the manuscript regarding neoplastic disease to primary versus metastatic disease. In this reviewer understanding, metastatic disease could be challenging regarding the differentiation with pneumonia. Is that correct?
Later in results, authors described primary lung tumors as primary diagnosis. Why primary lung tumors could be challenging regarding its diagnosis?
Please, divide the introduction section in different paragraphs
Do not start the results section with the word “Finally”.
Author Response
Reviewer: This manuscript proposes a diagram to differentiate lung neoplasia to pneumonia. However, authors mixed primary and metastatic diseases. Therefore, could be interesting to shown some x-ray images of the disease diseases, to increase reader understanding.
Our answer: done
Reviewer: Authors could provide more information in the manuscript to direct the manuscript regarding neoplastic disease to primary versus metastatic disease. In this reviewer understanding, metastatic disease could be challenging regarding the differentiation with pneumonia. Is that correct?
Our answer:
A metastatic neoplasm was diagnosed when a primary tumor was found in clinical examination or ultrasonography of the abdominal cavity. We added this information in the Materials and Methods section, lines 211-212. Hopefully, it is clearer now. Unfortunately, it is not possible to distinguish between primary and metastatic lesions based on the ultrasound image, because primary lesions are often not a single solid tumor, but in the course of a primary lung neoplasm, we observe all described changes, such as B lines, thickening of the pleural border, consolidation, hepatization.
Reviewer: Later in results, authors described primary lung tumors as primary diagnosis. Why primary lung tumors could be challenging regarding its diagnosis?
Our answer:
As we mentioned above, primary lung cancer is not only single lung tumors, but also various lesions, covering a large part of the lung.
Reviewer: Please, divide the introduction section in different paragraphs
Our answer: Done.
Reviewer: Do not start the results section with the word “Finally”.
Our answer: done.